# Determinants of COVID-19 vaccine uptake among persons with disabilities in three selected districts of Zambia

Allan Mayaba Mwiinde[1]*, Isaac Fwemba[2], Joseph Mumba Zulu[3,4], Choolwe Jacobs[1], Patrick Kaonga[1]

1 Department of Epidemiology and Biostatistics, School of Public Health, University of Zambia, Lusaka, Zambia, 2 Department of Medical Education, School of Medicine, University of Zambia, Lusaka, Zambia, 3 Department of Health Promotion, School of Public Health, University of Zambia, Lusaka, Zambia, 4 Department of Health Policy and Management, School of Public Health, Lusaka, University of Zambia, Lusaka, Zambia

* mayabamwiinde@gmail.com

## Abstract

COVID-19, is still a public health threat due to uncertainties around the potential evolution of the virus, changes in clinical characteristics, and the introduction of new prevention and therapeutic modalities. Persons with Disabilities (PWDs) were among the most highly affected groups by the COVID-19 pandemic due to their underlying conditions. This study aimed to establish the prevalence and identify the determinants of COVID-19 vaccine uptake among PWDs in three selected districts of Zambia. A cross-sectional study was conducted from June to August 2023 in Lusaka, Mazabuka, and Monze Districts. Structured questionnaires were developed and administered to PWDs aged 18 years and above. Log-binomial model was used to calculate the adjusted prevalence ratios (aPRs) and 95% confidence intervals (CIs) for assessing COVID-19 vaccine acceptance and its determinants. Out of the sample of 985, the proportion of COVID-19 vaccine uptake was 67.6% (95% CI: 65%, 71%). Vaccine uptake was higher among the married (aPR: 1.21; 95% CI:1.06, 1.39) participants from Mazabuka District (aPR: 1.61; 95% CI: 1.34, 1.93) living in rural areas (aPR:1.27; 95% CI: 1.09, 1.49) having extra income (aPR:1.27; 95% CI: 1.09,1.50) involved in routine health checkup (aPR:1.23; 95% CI: 1.11, 1.37) previously infected with COVID-19 (aPR:1.32; 95% CI, 1.04-1.68) previously vaccinated (aPR: 1.16; 95% CI: 1.00, 1.33) and understanding safety of the vaccine (aPR: 2.33; 95% CI: 1.55, 3.49). Conversely, low vaccine uptake was observed among participants earning less than k200 from social cash transfer (aPR: 0.79; 95% CI: 0.71, 0.87). More research is needed to identify determinants of vaccine uptake among PWDs that make them more vulnerable to infectious diseases such as COVID-19. There is need to improve vaccination coverage among PWDs. A more holistic and inclusive health promotion approach needs to be adopted to ensure that PWDs are not left behind in accessing vaccines.

**Data availability statement:** All relevant data are within the paper and its Supporting Information files.

**Funding:** This work was supported by NORHED-PRICE, Grant Number 70324 to AMM. The funder had no role in study design, data collection and analysis, decision to publish, or preparation of the manuscript

**Competing interests:** The authors have declared that no competing interests exist.

## Introduction

Coronavirus disease 2019 (COVID-19) is an infectious disease caused by the severe acute respiratory syndrome coronavirus 2 (SARS-CoV-2 virus) [1]. Although the pandemic has slowed down globally, there are still new COVID-19 cases being reported globally [2]. Between July 22 and August 18, 2024, there were 238,000 new cases recorded in 91 countries (39%) and 4400 deaths reported in 35 countries (15%). These figures represent an increase of 23% and 44%, respectively, over the prior period from June 24 to August 18, 2024 [2,3].

With the current ongoing infection of COVID-19, vaccines are key pharmaceutical interventions to contain COVID-19, and their development and acceleration without compromising safety have proved to provide strong protection against severe illness and death among humans [4,5]. Although a person can still become infected with COVID-19 after vaccination, they are more likely to have mild or no symptoms [6]. Most infected patients with COVID-19 are managed through supportive care [7]. COVID-19 vaccination remains one of the most cost-effective public health interventions with improved outcomes to help end the disease [8].

Currently, there are growing concerns about vaccine uptake, which are due to challenges in access, high cost of production and purchase price, and high levels of misinformation (infodemic) on the effectiveness and safety of the vaccines [9,10]. Learning from the COVID-19 pandemic, different strategies need to be developed to improve the uptake of vaccines among all age groups in order to maximise the potential of vaccination around the world [10]. According to World Health Organization (WHO) guidelines on improved uptake of vaccination, there is a need to address uptake at the country or sub-group level in order to develop an understanding of the magnitude and setting of the challenges that may reduce uptake [11].

Persons with Disabilities (PWDs) have proved to be facing challenges in accessing primary health care (PHC) services including access to vaccination compared to the general population [12]. This is particularly true for those residing in rural areas and are PWDs [12].

It is well known that many PWDs have underlying health conditions that increase the risk of poor outcomes if they contract diseases like COVID-19 [13]. Despite the available knowledge, governments have not been proactive in responding to the unique healthcare needs of PWDs before and during the COVID-19 pandemic [14].

Zambia recorded its first COVID-19 case on 20th March 2020 and launched the vaccination rollout on 14th April 2021 after receiving the first COVID-19 vaccine consignment on 5th March 2021 [15,16] . Initial vaccination was meant for high risk populations such as those with comorbid conditions such as diabetes, hypertension, older population, and healthcare workers [17]. In August 2021, the vaccine was rolled out to the public including PWDs aged 18 years and older [18]. The government ran campaigns through various forms of media to encourage uptake of the vaccine. Vaccines were accessible in various government health facilities and mobile vaccination points [19]. However, there was low uptake in the early days of vaccination due to misinformation and misconceptions regarding the vaccines [18]. To improve vaccination coverage and uptake, the government of the Republic of Zambia launched a new

vaccination strategy involving children from the age of 12–17 and adults 18 and above that would help reach the target of 70% vaccination of the entire population [20]. By November 2022 the national coverage of COVID-19 vaccination in Zambia was estimated to have reached the national target of 70% coverage [18].

In the adult population, studies that focused on vaccine acceptance estimated the coverage to be 72% among healthcare workers, while in the general population, acceptance was low in the early days of the pandemic [21,22]. Per vaccine uptake, parent's acceptance for COVID-19 vaccination was high for their children, however, they indicated high levels of hesitancy among themselves [23].

Currently, there are few studies in sub-Saharan African countries or Zambia inclusive that have established the prevalence and determinants of vaccine uptake, among PWDs [24]. Most of the studies that used the "health belief model" have focused on the general population, healthcare workers, low-income setting areas (slums), university students, and pregnant women [25–29]. Therefore, this study provides a unique perspective of using the health belief model to help understand the underpinning determinants in access to COVID-19 vaccines among PWDs. The lessons learnt will guide future vaccine policies and efforts to improve vaccine uptake and reduce hesitancy. In the past, vaccine hesitancy was high in Africa for COVID-19 and for other vaccines, like polio vaccine. If not addressed this is likely to sabotage the vaccination gains made already [30].

Therefore, this research aimed to establish the COVID-19 vaccination coverage and determinants of uptake among PWDs in three selected districts of Zambia.

## Methodology

### Study design, study area and study population

A cross-sectional study was conducted among PWDs in the Lusaka, Mazabuka, and Monze Districts of Zambia. The study was conducted from June to August 2023. Additionally, **Inclusion Criteria**

PWDs who are 18 years and above are registered with Social Welfare Department. Additionally, PWDs who were not on the social welfare register were included if they had an approved medical record of disability from a recognized government health institution.

### Exclusion criteria

Those who had stayed in Lusaka, Mazabuka, and Monze for less than six months were excluded.

**Operational definitions.** Disability was defined as a permanent physical, mental, intellectual, or sensory impairment that alone, or in combination with social or environmental barriers, hinders the ability of a person to fully or effectively participate in society on an equal basis with others [31].

COVID-19 vaccine uptake was defined as the proportion of the participants who were vaccinated with the COVID-19 vaccine when it was provided or available. Vaccine hesitancy was defined as the refusal or denial of uptake of the vaccine when it was readily available, as defined by the Strategic Advisory Group of Experts (SAGE) on Immunization [32].

### Study outcomes

The outcome variable of interest was uptake "ever received COVID-19 vaccine" which means the respondent has received or taken at least one dose of the COVID-19 vaccine. The second outcome variable of interest was no uptake of COVID-19 vaccine. This is further defined as respondents did not take a single dose of COVID-19.

### Sampling procedure

Two staged sampling procedures were conducted in this study. The three districts (Lusaka, Mazabuka and Monze) were selected using random sampling. Cities where the government had rolled out COVID-19 vaccination were classified into

one group and Lusaka was randomly selected. The towns that had high rural populations were classified into a second group where Mazabuka and Monze were selected. Secondly, a simple random selection process was used to select the participants in the study of about 985 PWDs. After being chosen at random and having their phone numbers listed on the Social Welfare Register, the participants were contacted at the closest meeting location and interviewed there. Individuals who were unable to come to the closest meeting location were contacted in the privacy of their own yards. The next available name was chosen from the sampling list and visited if the participants could not be contacted or their residences could not be located during the contact time. From each household, only one participant selected at random was included in the study even when they had two or more PWDs. Community Welfare Assistance Committee (CWACs) were fully involved at all stages in the community to help introduce the data collectors to the households and identify the households of PWD. The participants who were not on the social welfare list and had confirmed hospital records of disability were randomly selected in the households with the help of the CWACs. All participants who had psychological and cognitive disabilities were guided by their parents or guardians, who answered the questionnaire and signed the informed consent on their behalf.

## Sample size

The sample size was calculated using Cochrane´s formula for proportion [33]; considering a 5% margin of error, a vaccination prevalence of COVID-19 of 50% based on the Zambia National Public Health Institute (ZNPHI) dashboard on COVID-19 updates at 95% confidence interval [34]. A sample size of 985 was determined and proportion to size of each District was determined as indicated in the Table (S1 Table).

## Data collection tools and techniques

The structured questionnaire was used to collect the data based on the health belief model (HBM) which was used in this study to develop research questions on vaccination uptake (S1 Text) [33]. In this present study, the HBM was adopted as a conceptual flame work to assess PWD's determinants for vaccine uptake based on identified literature variables of interest (Fig 1). The HBM comprises mainly six key domains that impact a decision for vaccine uptake. Perceived susceptibility to COVID-19, perceived severity of COVID-19 infection, perceived benefits of COVID-19 vaccine, perceived barriers of

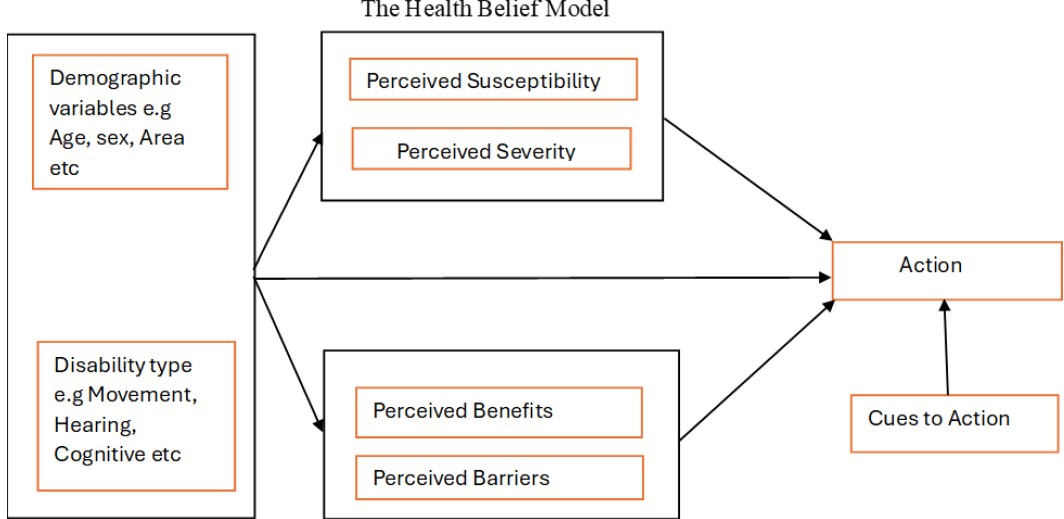

**Fig 1. The health belief Model Consonants used to obtain information from PWDs.**

COVID-19 vaccine, cues to action, and self-efficacy. Several individual-level disabilities and socio-demographic characteristics may also drive the uptake of the COVID-19 vaccines among PWDs [35].

The developed structured questionnaire was pre-tested outside the study districts to avoid influencing the data collection process in the districts of study (S1 Text). The purpose of pre-testing the questionnaire was to determine its validity, identify any ambiguous statements, and determine whether it would elicit the desired replies in order to achieve the objective. A sample size of about 30 participants was selected at random. The maximum time one questionnaire could take to administer was 17 minutes.

The structured questionnaire, which was closed-ended, was programmed into the Open Data Kit (ODK) collect version software which was used on the tablet-based application to administer the questionnaire in the three districts [36]. The final collected data in ODK was then converted into a Microsoft Excel spreadsheet (S1 Data).

### Statistical analysis

The data from Microsoft Excel was inputted into STATA version 17 software [37] for analysis. Categorical variables were summarised using frequencies and percentages. Descriptive analyses on the socio-demographic, state of disability age and gender characteristics of the participant's uptake of the COVID-19 vaccine, perceptions of the vaccine, self-reported determinants of their willingness to be vaccinated, and their trusted information sources and expectations about the vaccine were determined. The chi-square or Fisher exact test was conducted to assess the association between categorical variables on vaccination (uptake) versus socio-demographic characteristics and healthcare belief model consonants.

The Log-binomial regression model was used to assess the associated factors of COVID-19 vaccine uptake in this study among PWDs. The results were reported as prevalence ratios with their 95% confidence interval (CI) at a statistically significant level of 0.05. The unadjusted prevalence ratios and adjusted prevalence ratios were determined. The adjusted prevalence ratio was used to interpret the overall findings of the log-binomial regression model. The potential determinants used in the model were factors following the health belief model which included but were not limited to the following: age, sex, area of residence, district, urban or rural residence, education status, marital status, levels of severity of the disability, other determinants which were measured on the likelihood scale were perceived safety of the COVID-19 vaccine, perceived effectiveness of the COVID-19 vaccine, perceived risk of getting infected with COVID-19, perceived risk to members of the public, etc. The variables that were significant in chi-square analysis and bivariate analysis were included in the final log-binomial regression model. Furthermore, other variables that are highlighted in the literature that could influence vaccine uptake were also selected to be included in the final model. The best choice model was selected based on the investigator-led approach using the likelihood ratio test to compare the performance of the nested models.

## Results

### Demographic characteristics of participants (PWDs)

Of the total participants in the study (N = 985), 67.6% (n = 666/985) received the Covid-19 vaccine while the rest did not.

A total sample of 985 participants were interviewed with females comprising of 50.36% (n = 496/985) and the rest were males 49.64% (n = 489/985). The majority (37.9%, n = 373/985) were in the age group 18–35 years and not married (44%, n = 433). Most 45% (n = 443/985) reported primary level as their highest level of education and more than half (55%, n = 506/985) were from Lusaka District. More than three-quarters (88.60%, n = 872/985) were from urban areas and almost everyone (94.3%, n = 872/985) was unemployed. Additionally, half of the participants (51.4%, n = 506/985) were not receiving social cash transfers but of those who received, the majority 67.80% (n = 668/985) were receiving K400 per month. When the association between the independent variable and the outcome was assessed, age was associated with COVID-19 vaccine uptake (p = 0.007), marital status was associated with COVID-19 vaccine uptake (p = 0.002), education level was associated with COVID-19 vaccine uptake (p = 0.014), district was associated with COVID-19 vaccine uptake (p < 0.001), the region was associated with COVID-19 vaccine uptake (p = 0.001), social cash transfer was associated with

COVID-19 vaccine uptake (p = 0.001), the amount received on social cash transfer was associated with COVID-19 vaccine uptake (p < 0.001) (Table 1).

### Type of disability and the proportion of COVID-19 vaccine uptake

In the study, those who had taken the COVID-19 vaccine were PWDs with movement disability 51% (n = 555/985) spinal chord disability 14% (n = 133/985), head injuries 10% (n = 94/985), vision 17% (166/985), hearing 11% (n = 104/985), speech 11% (n = 112/985), cognitive 11 (n = 105/985), psychological 10 (n = 96/985) and invisible disabilities 10% (n = 96/985) (Fig 2).

### Proportion of disability by sex and location among PWDs

A total of 75.13% (n = 740/985) of participants had acquired disability. The total number of males were 49.32% (n = 365/740). Congenital disabilities accounted for 24.87% (n = 245/985) of this number males accounted for 50.6% (n = 124/245). A total of 55% (n = 543/985) participants were from Lusaka District, Mazabuka District had 20.1% (n = 205/985) and Monze District 24% (n = 237/985) (Fig 3). The highest disability was mobility with a total of 74% (n = 730/985), among this number males were 48% (Fig 3).

### Health belief model constructs influencing vaccination uptake among PWDs

**Past behaviour towards vaccination.** The total proportion of those who were vaccinated before as far as they can remember was 71.90% (n = 708/985), (95%, CI; 68.99-74.60). Those who indicated to have been previously vaccinated among the females were 50.71% (n = 359/708) and among males 49.29% (n = 349/708). COVID-19 vaccine was significantly higher (p < 0.001) among participants who were previously vaccinated (79.43%, n = 529/666) compared to those who have not been previously vaccinated (20.57%, n = 137/666).

**Perceived susceptibility.** Vaccine uptake was significantly higher (p = 0.004) among participants who were aware of or had heard of the COVID-19 pandemic (98.07%, n = 659/666) compared to those who were not aware of the COVID-19 pandemic (1.05%, n = 7/666). Uptake of the COVID-19 vaccine among those who considered COVID-19 disease to have high personal risk (45.35%, n = 302/666) was higher (p = <0.001) compared to those who considered COVID-19 disease to have row personal risk (27.93%, n = 186/666). Conversely, uptake of COVID-19 vaccine among respondents who indicated that COVID-19 disease had no risk to the public (52.25%, n = 348/666) was significantly (p = 0.006) higher compared to the respondents who indicated that COVID-19 disease posed a risk to the public (24.77%, n = 165/666) (S1 Data).

**Perceived severity of COVID-19.** Uptake of COVID-19 vaccine was significantly higher (p = <0.001) among participants who perceived that they are not likely to be infected with COVID-19 disease in future (34.74%, n = 231/666) compared to those who indicated that they are likely to be infected in future (27.82%, n = 185/666). Those without experience of seeing a COVID-19 patient or family member infected with COVID-19 (59.31%, n = 395/666) had higher COVID-19 vaccine uptake (p = 0.037) compared to those with no experience of seeing or having a COVID-19 patient at the household level (40.61%, 271/666). There was significantly higher (p = 0.017) COVID-19 vaccine uptake among those who have never been infected with COVID-19 disease before (96.40%, n = 642/666) compared to those who have been infected with COVID-19 disease (3.60%, n = 24/666). Vaccine uptake was significantly higher (p = 0.002) among participants who strongly disagreed that COVID-19 vaccine could cause disease and death (15.64%, n = 104/666) compared to participants who agreed that COVID-19 disease could cause illness and death (1.65%, n = 11/666) (S2 Table).

**Perceived benefits.** COVID-19 vaccine uptake was significantly higher (p = <0.001) among respondents who cited the COVID-19 vaccine's safety as justification for vaccination (92.4%, n = 651/666) compared to those who did not agree (2.25%, n = 15/666). COVID-19 vaccine uptake was significantly higher (p = <0.001) among participants who cited the effectiveness of the COVID-19 vaccine (93.52%, n = 621/666) compared to those who were not sure (5.42%, n = 36/666). COVID-19 vaccine uptake was significantly higher (<0.001) among participants who agreed that the vaccine is meant to protect (98.94%, n = 655/666) compared to those who disagreed (1.06%, n = 7/666) (S2 Table).

**Table 1. Association of demographic characteristics with COVID-19 uptake among PWDs in the three districts of Zambia.**

| Variable | Vaccinated n=666 (67.61%) | Un-vaccinated n=319 (%) | Total N=985 (100%) | P-value |
|---|---|---|---|---|
| **Age (years)** | | | | |
| 18-35 | 230 (34.53) | 143 (44.83) | 373 (37.87) | |
| 36-55 | 258 (38.74) | 100 (31.35) | 358 (36.35) | 0.007[c] |
| >55 | 178 (26.73) | 76 (23.82) | 254 (25.79) | |
| **Sex** | | | | |
| Male | 322 (48.35) | 167 (52.35) | 489 (49.64) | 0.240[c] |
| Female | 344 (51.65) | 152 (47.65) | 496 (50.36) | |
| **Marital Status** | | | | |
| Single | 267 (40.09) | 168 (52.66) | 435 (44.16) | |
| Married | 212 (31.83) | 74 (23.20) | 286 (29.04) | 0.002[c] |
| Divorced | 55 (8.26) | 23 (7.21) | 78 (7.92) | |
| Widowed | 132 (19.86) | 54 (16.93) | 186 (18.9) | |
| **Education** | | | | |
| Unschooled | 83 (12.46) | 54 (16.93) | 137 (13.91) | |
| Primary | 292 (43.84) | 149 (46.71) | 441 (44.77) | |
| Secondary | 265 (39.79) | 97 (30.41) | 362 (36.75) | 0.01[c] |
| Tertiary | 26 (3.90) | 19 (5.96) | 45 (4.57) | |
| **District** | | | | |
| Lusaka | 327 (49.10) | 216 (67.71) | 543 (55.13) | <0.001[c] |
| Mazabuka | 163 (24.47) | 42 (13.17) | 205 (20.81) | |
| Monze | 176 (26.43) | 61 (19.12) | 237 (24.06) | |
| **Denomination attended** | | | | |
| Protestants | 376 (56.46) | 182 (57.23) | 558 (56.71) | 0.743 |
| Pentecostals | 151 (22.67) | 76 (23.90) | 227 (23.07) | |
| Catholics | 139 (20.87) | 60 (18.87) | 199 (20.22) | |
| **Region** | | | | |
| Urban | 572 (85.89) | 292 (92.99) | 869 (88.22) | |
| Rural | 94 (14.11) | 22 (7.01) | 116 (11.78) | 0.001[c] |
| **Employment status** | | | | |
| Unemployed | 627 (94.14) | 302 (94.67) | 929 (94.31) | |
| Informal Employment | 36 (5.41) | 15 (4.70) | 51 (5.18) | 0.766[f] |
| Formal Employment | 3 (0.30) | 2 (0.63) | 5 (0.50) | |
| **Social Cash Support (ZMWK)[e]** | | | | |
| 200 | 112 (16.82) | 25 (7.84) | 137 (13.91) | |
| >200 | 554 (83.18) | 294 (92.16) | 848 (86.09) | <0.001[f] |
| **Other Income (ZMWK)[e]** | | | | |
| 200 | 634 (95.20) | 312 (31.68) | 946 (96.04) | |
| >400 | 32 (4.80) | 3 (0.30) | 39 (3.96) | 0.049 [c] |

1. c=Chi-square test, f=fisher's exact test. e=Exchange rate, Zambian Kwacha (ZMW) 231– US Dollar, Social Cash Support=Income received from Government support through social cash transfer, Other Income=Income earned through work.

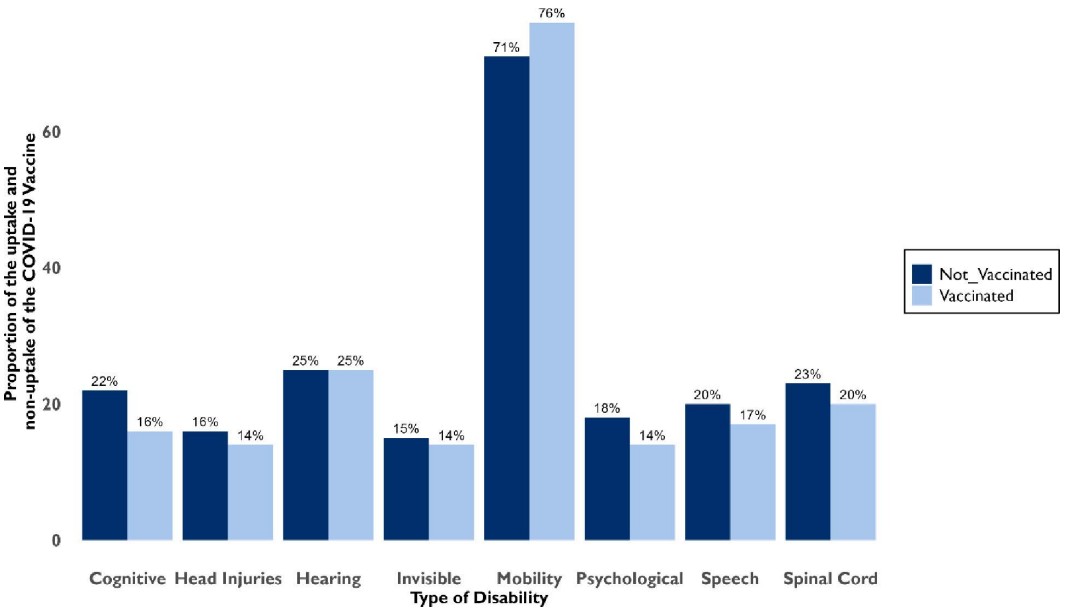

**Fig 2. Type of disability and the proportion of the COVID-19 vaccine uptake among PWDs.**

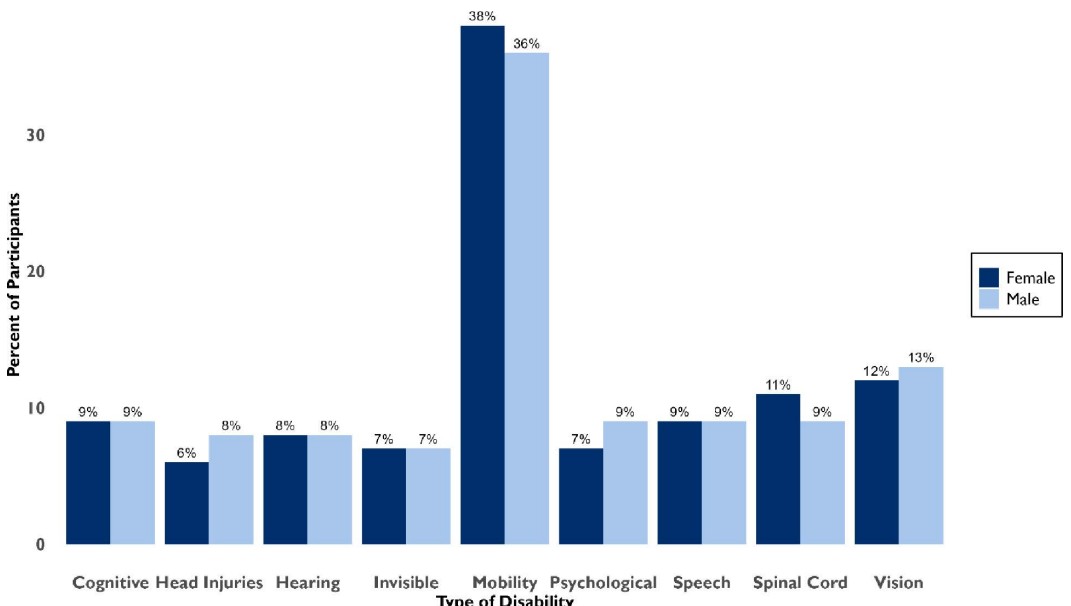

**Fig 3. The proportion of Disability by sex among PWDs.**

**Perceived barriers.** COVID-19 vaccine uptake was significantly higher (p=<0.001) among participants who have been vaccinated before (79.43%, n = 529/666) compared to those who have never been vaccinated before as far as they can remember (20.57%, n = 137/666). Uptake of the COVID-19 vaccine was significantly higher (p = 0.002) among participants who disagreed that COVID-19 vaccine causes fertility challenges (48.34%, n = 320/666) compared to those who agreed (0.91%, n = 6/666) (S2 Table).

About half of vaccinated participants who indicated having the ability to visit PHC on their own had significantly higher uptake (p = 0.018) of the COVID-19 vaccine (49.40%, n = 666) compared to those who did not have the capacity to visits the PHC on their own (50.60%, n = 337/666). COVID-19 vaccine uptake was significantly higher (p < 0.001) among participants who visited the PHC for routine health checkups (75.70%, n = 408/666) compared to those who did not visit the PHC facilities for screening. Among participants who received social support there was significantly higher uptake (p = 0.001) of COVID-19 vaccine (54.95%, n = 366) compared to those who did not receive social support (45.05%, n-300/666). COVID-19 vaccine uptake was higher (p = 0.013) among participants who received ZMWK 400 from social support (64.84%, n = 236/666) than those who earned ZMWK 200 (30.77%, n-112/666). Those who earned less than a ZMWK 200 a month from their labour had significantly higher uptake (p = 0.049) of the COVID-19 vaccine (95.20%, n = 634/666) than those who earned more than ZMWK 400 (4.80%, n = 32/666). There was a significant higher uptake (p = 0.001) among participants who received health education on COVID-19 disease and vaccines by health workers (91.43%, n = 608)compared to those who indicated that they never received health education by a health worker on COVID-19 disease and COVID-19 vaccines (8.57%, n = 57/666) (S2 Table).

**Ques to action**

Among the participants (4.65%, n = 31/666) were vaccinated because they were directly told by the government worker compared to those who indicated that they did so in order to protect others from COVID-19 (11.11%, n = 74/666).

Participants who accepted the existence of COVID-19 disease pandemic (28.08%, n = 187/666) had significantly higher uptake (p = 0.002) compared to those who disagreed (1.35%, n = 9/666) (S2 Table).

**Determinants of COVID-19 vaccine uptake in log-binomial regression model**

The results of the binary log-binomial regression model provided the unadjusted prevalence ratio (aPR) and the adjusted prevalence ratio (aPR) for the log-binomial regression model (Table 2). Respondents who were married had 1.21 higher prevalence of uptake of the COVID-19 vaccine than those who were single (aPR = 1.21; 95% Confidence Interval (CI) 1.06, 1.39), holding all factors constant. Respondents who were widowed had 1.29 higher prevalence of uptake of the COVID-19 vaccine than those who were single (aPR: 1.29, 95% CI: 1.08-1.53).

Participants from Mazabuka District had 1.61 higher prevalence of uptake compared to their counterparts from Lusaka District (aPR = 1.61, 95% CI: 1.34, 1.93) holding all factors constant. Respondents from rural areas had 1.27 higher prevalence of uptake of the COVID-19 vaccine than those in the urban areas (aPR: 1.27, 95% CI: 1.09-1.49).

Respondents with extra sources of income more than K200-00 had higher prevalence of 1.27 of uptake of the COVID-19 vaccine than those who did not have extra sources of income (aPR = 1.27, 95% CI:1.09, 1.50).

Participants who visited the PHC facilities for routine health check up had 1.23 higher prevalence of uptake of the COVID-19 vaccine than those who did not visit the PHC facility for their routine health checks (aPR:1.23, 95% CI:1.11, 1.37). Those who indicated that they have been infected with COVID-19 before had 1.32 higher prevalence of COVID-19 vaccine uptake than those who have never been infected before (aPR: 1.32, 95% CI: 1.04, 1.68). Participants who were confident that they would not be infected with COVID-19 in future had 2.28 higher prevalence of COVID-19 vaccine uptake (aPR: 2.28, 95% CI: 1.00, 1.36).

When asked if they had ever been vaccinated before, those who were previously vaccinated had higher prevalence of 1.16 of uptake than those who were not previously vaccinated (aPR:1.16, 95% CI: 1.00, 1.33). When asked about the safety of the vaccine, those who agreed that the vaccine was safe had 2.33 higher prevalence of uptake of the COVID-19 vaccine than those who disagreed (aPR: 2.33, 95% CI: 1.55-3.49) (Table 2).

Participants who received money as social cash transfer amount less than K200-00 had 79% lower prevalence of COVID-19 vaccine uptake than those who received more than k200-00 (aPR: 0.79, 95% CI: 0.71, 0.87).

**Table 2. Determinants of vaccine uptake among PWDs.**

| Variable | Category | Un Adjusted PR (95, CI) | P-Value | Adjusted PR (95%, CI) | P-Value |
|---|---|---|---|---|---|
| **Age (Years)** | 18-35 | Ref | | Ref | |
| | 36-55 | 1.16 (1.05-1.30) | 0.003 | 1.02 (0.90-1.09) | 0.783 |
| | >56 | 1.13 (1.01-1.27) | 0.027 | 0.89 (0.79-1.05) | 0.166 |
| **Sex** | Female | Ref | | Ref | |
| | Male | 0.94 (0.87-1.04) | 0.240 | 0.99 (0.90-1.09) | 0.783 |
| **Marital Status** | Single | Ref | | Ref | |
| | Married | 1.21 (1.09-1.34) | <0.001 | 1.21 (1.06-1.39) | **0.006** |
| | Divorced | 1.15 (0.97-1.35) | 0.093 | 0.89 (0.74-1.06) | 0.200 |
| | Widowed | 1.16 (1.03-1.30) | 0.016 | 1.29 (1.08-1.53) | **0.005** |
| **Educations Status** | Unschooled | Ref | | Ref | |
| | Primary | 1.09 (0.94-1.27) | 0.248 | 0.95 (0.83- 1.10) | 0.509 |
| | Secondary | 1.21 (1.04-1.40) | 0.013 | 1.06 (0.91-1.23) | 0.445 |
| | Tertiary | 0.95 (0.72- 127) | 0.743 | 0.80 (0.63-1.03) | 0.078 |
| **District** | Lusaka | Ref | | Ref | |
| | Mazabuka | 1.32 (1.20-1.46) | <0.001 | 1.61 (1.34-1.93) | **<0.001** |
| | Monze | 1.23 (1.12-1.36) | <0.001 | 1.10 (0.91-1.32) | 0.316 |
| **Area** | Urban | Ref | | Ref | |
| | Rural | 1.23 (1.11-1.36) | <0.001 | 1.27 (1.09-1.49) | **0.003** |
| **Amount Received on Social Cash** | ≤ K200 | Ref | | Ref | |
| | >K200 | 0.79 (0.73-0.88) | <0.001 | 0.79 (0.71-0.87) | **<0.001** |
| | | | | | |
| **Amount Earned** | ≤ 200 | | | | |
| | >200 | 1.23 (1.05-1.43) | 0.010 | 1.27 (1.09-1.50) | **0.003** |
| **Capacity to visit PHC facility** | No (Ref) | | | | |
| | Yes | 1.11 (1.02-1.21) | 0.018 | 1.03 (0.94-1.13) | 0.464 |
| **Visits PHC for health checks** | No | Ref | | | |
| | Yes | 1.31 (1.19-1.44) | <0.001 | 1.23 (1.11-1.37) | **<0.001** |
| **Infected with COVID-19 Before** | No | Ref | | Ref | |
| | Yes | 1.33 (1.15-1.53) | <0.001 | 1.32 (1.04-1.68) | **0.023** |
| **Are you likely to be infected in the future** | Strongly agree | Ref | | | |
| | Agree | 1.14 (1.00-1.30) | 0.044 | 1.01 (0.88-1.17) | 0.838 |
| | Disagree | 1.31 (1.16-1.47) | <0.001 | 2.28 (1.00-2.56) | **0.045** |
| | Strongly Disagree | 1.23 (1.10-138) | <0.001 | 1.05 (0.83- 1-32) | 0.711 |
| **COVID-19 a Risk to People** | No Risk | Ref | | | |
| | Minor Risk | 0.93 (0.81-1.06) | 0.279 | 0.87 (0.74-1.03) | 0.103 |
| | Not sure | 74 (0.59-0.93) | 0.010 | 1.03 (0.84-1.27) | 0. 761 |
| | Moderate Risk | 80 (0.63-1.01) | 0.071 | 0.81 (0.66-0.99) | **0.043** |
| | High Risk | 1.00 (0.91-1.11) | 0.887 | 0.68 (0.54-0.87) | **0.002** |
| **Seen COVID-19 Patient** | No | Ref | | Ref | |
| | Yes | 1.10 (1.00-1.20) | 0.033 | 1.04 (0.92-1.18) | 0.504 |
| **Ever been vaccinated before** | No | Ref | | Ref | |
| | Yes | 1.51 (1.33-1.71) | <0.001 | 1.16 (1.00-1.33) | **0.038** |
| **Safety of Vaccines** | No | Ref | | Ref | |
| | Yes | 3.53 (2.24-5.56) | <0.001 | 2.33 (1.55-3.49) | **<0.001** |
| **Vaccines are Protective** | No | Ref | | Ref | |
| | Yes | 0.48 (0.26-0.87) | 0.017 | 0.66 (0.39-1.13) | 0.128 |

*(Continued)*

**Table 2.** (Continued)

| Variable | Category | Un Adjusted PR (95, CI) | P-Value | Adjusted PR (95%, CI) | P-Value |
|---|---|---|---|---|---|
| **COVID-19 Human caused** | Strongly agree | Ref | | | |
| | Agree | 1.(0.82-1.47) | 0.527 | 1.01 (0.79-1.29) | 0.932 |
| | Not Sure | 1.16 (0.88-1.54) | 0.295 | 0.94 (0.78-1.15) | 0.578 |
| | Disagree | 1.19 (0.90-1.57) | 0.227 | 1.19 (0.96-1.48) | 0.116 |
| | Strongly Disagree | 1.29 (0.96-1.72) | 0.089 | 0.99 (0.75-1.30) | 0.940 |

PR=Prevalence Ratio, aPR= Adjusted Prevalence Ratio, CI= Confidence Interval

When asked about the Risk of COVID-19 Participants who indicated that COVID-19 had a moderate risk had 81% lower prevalence of COVID-19 vaccination compared to those who indicated that there is low risk (Table 2).

## Discussion

To the best of our knowledge, this is the first study in Zambia to assess the COVID-19 vaccine uptake and their determinants among PWDs. The need to investigate the factors influencing vaccine uptake is of great importance to the global community with the current emerging and re-emerging infectious diseases taking centre stage. This study established that the proportion of the PWDs vaccinated was at 67.6%. The low levels of vaccination among PWDs in Zambia follow a similar pattern to other sub-Saharan African countries by not reaching the required herd immunity of 80% [38–40]. The proportion (67.6%) among the PWDs could be attributed to the fact that the caregivers have a role to play in helping disseminate messages of vaccination among the PWDs. However, it must be noted that the determinants of uptake among PWDs established in the study such as marriage, district, previous history of vaccination, and the safety of the vaccine among others, are similar to what other studies have established in the general population except that among PWDs their state of disability plays a critical role in influencing certain outcomes, such as those who visit the health facility for their routine health checks are more likely to be vaccinated [41–43]. Following the proportions of vaccination at the district level Lusaka had the lowest levels of vaccine uptake compared to Mazabuka and Monze. Furthermore, the determinants of uptake were higher in Mazabuka than in Lusaka the capital of Zambia. Similar to the findings of Carcelin et al., [23] one plausible reason could be that the urban populations have access to the internet and social media which influences less uptake, and this may have impacted their beliefs on COVID-19 vaccines. In our study, Mazabuka and Monze Districts with rural populations compared to Lusaka have limited access to social media and the internet resulting in less misinformation and misconception spreading around and therefore more uptake of the vaccine. In sub-Saharan African countries, social media has been the prominent driver of COVID-19 vaccine low uptake [44]. These findings are consistent with the findings of Wu et al., [45] who also indicated that the likelihood of being vaccinated in rural areas was higher than in urban areas due to a lack of trust in the pharmaceutical industries producing the COVID-19 vaccine.

Our results also shed light on the difficulties PWDs have getting vaccinated because of the levels of disability. Critical to our findings is the side income earned apart from the social cash transfer provided by the government of the Republic of Zambia. The severity levels of disability among PWDs may also make it impossible for other PWDs to participate in active work or be exposed to the ongoing vaccination campaigns, as those who reported having extra income were more likely to be vaccinated than those who did not. This can be attributed to the communication barriers of not receiving timely information, unable to respond effectively to changes in regulations where instruction leaflets in Braille are not provided for persons with vision disabilities [46–48]. Those who have spouses or are married are more likely to be vaccinated than those who do not have spouses. This indicates the inadequacies of social support in care or the lack of caregivers at the community/household level. The findings are similar to those of Liu et al., [49] in the abled population, showing that those who were married were more likely to receive vaccination than those who were single or divorced. However, our findings

also indicate that those who were widowed were more likely to be vaccinated than those who were single, this could be attributed to the availability of care givers such as children who can support the parents.

Our findings further reveal that among PWDs vaccine uptake was associated with understanding that the vaccine was safe for the general population and those who had low levels of uptake could only attribute it to its ineffectiveness in preventing COVID-19 disease. This is different from the findings of Myers et al [50] who indicated that in the USA, PWDs were only concerned with the issue of safety of the vaccine. This can be attributed to misinformation spread on social media platforms which has caused less uptake among members of the public. To manage an infodemic effectively, a new public health taxonomy for social listening on respiratory pathogens has been developed [51,52]. False claims about adverse vaccine side effects, such as vaccines being the cause of autism, were already considered a threat to global health before the outbreak of COVID-19. These findings by Skafle et al., [53] have the potential to cause low levels of vaccine uptake among PWDs for fear of worsening their situation as indicated in our findings of the perceived barriers of COVID-19 vaccination among PWDs.

Furthermore, the sensitization of PWDs in understanding the importance of COVID-19 vaccine was one of the most important determinants of vaccine uptake to those who indicated having less likelihood of being infected in future [54]. This indicates the need to effectively provide the much-needed extensive health education among this vulnerable group as indicated by Lohiniva et al., [55] that individual risk perception is not only linked with individual factors but also with broader sociocultural values. This outcome can be attributed to those who indicated that they could recall when they were vaccinated and were more associated with vaccine uptake than those who indicated that they could not recall.

The study also established that those who were vaccinated before were more likely to uptake the COVID-19 vaccine than those who did not recall having taken any other vaccines. These findings are similar to what Belay et al., [56] established that in adulthood previous immunization histories were one of the key determinants of influencing vaccination in Ethiopia. This suggests that involving PWDs in ongoing vaccination efforts before the occurrence of a pandemic would enhance uptake during global health crises, such as COVID-19. The results also indicate that those who were infected before were more likely to uptake the COVID-19 vaccine similar to the findings of Samarasekera, 2021 who indicated that someone who tested positive was more likely to be vaccinated than those who were not infected [57].

Those who perceived that COVID-19 had a moderate risk to members of the public were associated with the likelihood of less uptake. This is most likely due to the COVID-19 conspiracy theories that have demonstrated the negative pandemic-related behaviour in the abled population indicating that among PWDs they have also been negatively affected. Due to their confidence in the vaccine and knowledge of the protection it produces, participants who disagreed that they were unlikely to be infected by COVID-19 in the future were more likely to get vaccinated than participants who agreed that they could be infected. However, there is need to conduct studies on the immunity of an individual after being vaccinated from COVID-19 to provide education when an individual can seek to have another COVID-19 vaccine uptake. Available literature has only indicated that getting a vaccine will only boost immunity or provide added protection for a certain period of time [58]. Further information is required to provide guidance to health education activities on how long the COVID-19 vaccine will provide the expected prolonged immunity before one seeks to uptake another vaccine dose. Furthermore, our findings reveal that there is still a gap that exists in reaching PWDs with health vaccination messages during pandemics that are due to speech and hearing impairment. Similar to the findings in USA there is a challenge in information dissemination among PWDs which affects access to health services such as vaccination as a result disability perspectives are important during the pandemic [59]. The challenges causing inequality also exist in developed countries, PWD reported worse access to healthcare, due to transportation costs, and distance to the primary healthcare facilities [60].

Therefore, government efforts towards reaching PWDs should be enhanced to have an inclusive and resilient health system that would withstand the shocks of emerging diseases such as COVID-19. This is similar to the recommendations of WHO, [61,62] that information and communication in diverse formats to suit the different needs of PWDs should be delivered without relying much on either spoken or written information to strengthen PHC.

National healthcare system capacity needs to be strengthened to manage infectious diseases such as COVID-19 [63]. The COVID-19 vaccination services must be integrated with other immunization services and alongside other health and social interventions to have a meaningful impact and to build long-term capacity in achieving herd immunity [64]. However, there is still a problem in most of the sub-Saharan African countries with reduced uptake and refusal to be vaccinated, these challenges must be addressed at community and individual levels [22]. To help address the inequalities among vulnerable groups such as PWDs, strengthening the PHC system through improved vaccine uptake while understanding factors they are influenced such as cultural ethnicity, policy, and environment would be of great importance [32,65].

In order to boost vaccine uptake before, during, and after pandemics, understanding the determinants of vaccine uptake among PWDs will help policymakers prioritise the necessary interventions. As vaccination rollout continues in LMICs there is a need to ensure that the determinants of vaccine uptake are enhanced, and determinants associated with less uptake are reduced among vulnerable groups if the global agenda of health for all 2030 is to be archived. The significance of the health belief model in the context of vaccine uptake helps to identify the individual determinants among PWDs that need to be enhanced to improve health promotion and disease prevention as seen in the context of our findings.

## Limitations of the study

This research employed the cross-sectional study design which could not establish the cause and effect relationship between independent factors and the outcome. Furthermore, participants' responses may be influenced by the desire to be provided with solutions to the challenges they face as PWDs. Using open-ended questions would have led to the development of themes to guide the discussions better. There is need to conduct a country wide survey to have a true understanding of the determinants influencing vaccination among PWDs in Zambia.

## Conclusion

More research is needed to identify determinants of vaccine uptake among PWDs that make them more vulnerable to infectious diseases such as COVID-19 to avoid adverse outcomes that may complicate their health outcome or death.

Improved vaccination coverage among vulnerable populations such as PWDs will contribute to the prevention of vaccine preventable infectious diseases. Special messages targeting PWDs should be developed to ensure that they are not left behind. Further studies on risk perception should be developed to ensure inclusiveness in all health matters among PWDs. There is need to develop policy strategies that address access to vaccines among PWDs with attention to the level of disability among this vulnerable group in Zambia. This will help to improve access to PHC and indirectly contribute towards the attainment of universal health coverage.

## Supporting information

**S1 Table. The sample size determination.**
(DOCX)

**S2 Table. Self-Reported determinants of the health belief model consonants and COVID-19 uptake among PWDs.**
(DOCX)

**S1 Text.**
(PDF)

**S1 Data.**
(XLSX)

## Author contributions

**Conceptualization:** Allan Mayaba Mwiinde.

**Data curation:** Allan Mayaba Mwiinde.

**Formal analysis:** Allan Mayaba Mwiinde.

**Funding acquisition:** Allan Mayaba Mwiinde, Joseph Mumba Zulu, Patrick Kaonga.

**Investigation:** Allan Mayaba Mwiinde.

**Methodology:** Allan Mayaba Mwiinde.

**Project administration:** Allan Mayaba Mwiinde, Joseph Mumba Zulu, Choolwe Jacobs.

**Resources:** Allan Mayaba Mwiinde, Joseph Mumba Zulu, Choolwe Jacobs.

**Software:** Allan Mayaba Mwiinde.

**Supervision:** Allan Mayaba Mwiinde, Isaac Fwemba, Joseph Mumba Zulu, Choolwe Jacobs, Patrick Kaonga.

**Validation:** Allan Mayaba Mwiinde, Isaac Fwemba, Joseph Mumba Zulu, Patrick Kaonga.

**Visualization:** Allan Mayaba Mwiinde, Isaac Fwemba, Joseph Mumba Zulu, Choolwe Jacobs, Patrick Kaonga.

**Writing – original draft:** Allan Mayaba Mwiinde, Isaac Fwemba, Joseph Mumba Zulu, Choolwe Jacobs, Patrick Kaonga.

**Writing – review & editing:** Allan Mayaba Mwiinde, Isaac Fwemba, Joseph Mumba Zulu, Choolwe Jacobs, Patrick Kaonga.

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
