## [Decision Letter · Decision Letter 0]

PGPH-D-24-01701

Determinants of COVID-19 Vaccine Uptake Among People with Disabilities  in Three Selected Districts of Zambia

Dear Allan,

Thank you for submitting your manuscript to PLOS Global Public Health. After careful consideration, we feel that it has merit but does not fully meet PLOS Global Public Health’s publication criteria as it currently stands. Therefore, we invite you to submit a revised version of the manuscript that addresses the points raised during the review process.

We look forward to receiving your revised manuscript.

Kind regards,

Collins Otieno Asweto, PhD

Academic Editor

Reviewer's Responses to Questions

**Comments to the Author**

1. Does this manuscript meet PLOS Global Public Health’s publication criteria?

Reviewer #1: Partly

Reviewer #2: Partly

Reviewer #3: Yes

2. Has the statistical analysis been performed appropriately and rigorously?

Reviewer #1: No

Reviewer #2: I don't know

Reviewer #3: Yes

3. Have the authors made all data underlying the findings in their manuscript fully available (please refer to the Data Availability Statement at the start of the manuscript PDF file)?

Reviewer #1: Yes

Reviewer #2: Yes

Reviewer #3: Yes

4. Is the manuscript presented in an intelligible fashion and written in standard English?

Reviewer #1: No

Reviewer #2: Yes

Reviewer #3: Yes

Reviewer #1: The study focuses on COVID-19 vaccine uptake among persons with disabilities (PWDs) and uses the Health Belief Model (HBM) as a conceptual framework to assess determinants of vaccine acceptance. The study provides valuable insights into the factors influencing COVID-19 vaccine uptake among PWDs. However, the manuscript would benefit from improved organization and clearer, more concise presentation of the results. Below are detailed suggestions for improvements:

Introduction

• Line 57-59: Comparing the share of the world population between the USA, the European Union, and Africa with their respective vaccination coverage will provide a clearer illustration of the global disparities in vaccine distribution. (Africa: 17.5% of the world population, 1.63% vaccination coverage by June 2021; Europe 9.6% of the world population and 18% of vaccine coverage… Tatar M, et al. COVID-19 vaccine inequality: A global perspective. J Glob Health. 2022 Oct 14;12:03072. doi: 10.7189/jogh.12.03072.)

• PWD. Line 66 spell out all the abbreviation, the first time it is used in the body of the manuscript, even if you do introduce it in the abstract.

• Line 68 do you mean inequality or inequity?

• Line 71 Spell out PHC, you don’t use it again, no need to abbreviate.

• Line 71-73: this sentence is confusing, do you mean: “To help address the inequalities faced by vulnerable groups such as PWDs, strengthening the primary health care system by improving vaccine acceptance and understanding the influencing factors such as cultural background, policy, and environmental conditions would be of great importance”.

• Line 80-81: I would rephrase it : “It is well known that many PWDs have underlying health conditions that increase their risk of poor outcomes if they contract diseases like COVID-19.”

• Line 85: Please spell out “SDGs”

Methods

• Line 123 Study outcomes:

o The objectives of this study were to establish the prevalence and determinants of COVID-19 vaccine 98 uptake among PWDs in three selected districts of Zambia. The outcomes only mentioned uptake of vaccine, what about the determinants?

o Vaccination prevalence is a confusing terminology, do you mean vaccination coverage?

o The study defines the primary outcome as binary (vaccine uptake vs. no uptake), which may oversimplify the behavior. A more nuanced outcome measure (e.g., partial uptake, intention to vaccinate, number of vaccine doses taken...) could provide a better understanding of the barriers and facilitators of vaccine uptake among PWDs.

• Could you include a paragraph on the ethical considerations of the study? Specifically, was the study approved by a local Institutional Review Board (IRB)? How was informed consent obtained from participants, especially those with cognitive disabilities? Additionally, how is the confidentiality of participant data being managed?

• Line 195: Only participants over 18 were included in this study, but here you mentioned adolescents, please clarify.

• Line 192 statistical analysis:

o While the study uses HBM constructs, the analysis might not fully explore the complex interrelationships between these constructs. The description suggests that only basic statistical tests (chi-square, Fisher exact test) were used to assess associations, without deeper analysis (e.g., interaction effects, mediation analysis) that could provide more insights into the factors influencing vaccine uptake.

o Although the study adjusted for some variables in the logistic regression model, the selection of variables appears to be based on significance in bivariate analysis, which might overlook potential confounders not highlighted in the literature. This could result in omitted variable bias.

o Line 205: It’s not necessary to include how you coded your data for analysis in your methods.

Results

• Line 227-234: The association between the independent variables and the outcome can be presented better. The study reports numerous statistically significant associations between various factors and vaccine uptake. However, there may be an overemphasis on p-values rather than considering the practical significance or effect sizes of these associations. For example, while some factors may be statistically significant, their actual impact on vaccine uptake might be minimal.

• Line 264: While the study measures several HBM constructs (such as perceived susceptibility, severity, and benefits), it appears that not all constructs were explored with equal depth. For instance, perceived barriers to vaccination, which are critical in understanding vaccine hesitancy, were only partially addressed, focusing mainly on misconceptions like the vaccine causing fertility issues or illness and death. However, other barriers such as access to vaccination sites, social support, or communication barriers were not deeply explored.

• Line 305: The study identifies several associations between sociodemographic factors (e.g., age, marital status, district, region) and vaccine uptake. However, it’s unclear whether these associations were fully adjusted for potential confounding variables, such as access to healthcare, education about vaccines, or disability-related challenges (e.g., mobility issues that may affect access to vaccination centers).

While the study provides valuable insights into the factors influencing COVID-19 vaccine uptake among PWDs, there are several weaknesses when comparing the results to the methods, particularly in the context of the HBM. The inconsistent application of HBM constructs, reliance on self-reported data, potential confounding factors, overemphasis on statistical significance, and generalizability issues all contribute to limitations in the study's findings.

Reviewer #2: AS from study findings coverage of vaccination is almost equal with general population why there is need to increase coverage to PWD ?

A more holistic and inclusive health promotion approach needs to be adopted to ensure that vulnerable communities are not left behind in accessing vaccines. how you can justify and why it should in the abstract.

What is the meaning of pandemic slow down as who has already declared over?

give justification of the need of this research at this stage in your country where the coverage almost equal with general population

Add information about country health sysytem of zambia on routine immunization coverage in introduction parDoes Does this study has scope to comments on linkage beetween Covid 19 and SDG goal? justify

What new evidence you want to add in existing context ?

Where is the comparision beetwen national covid 19 vaccnation coverage with this findings? and why you choose conviniently 3 district as compared to others districs give justification?

Add this finding in methodology and compare in discussion section

in table 1 aggregate total column % is more than 100 ?

How can you make a conclusion that There is need to develop bottom-up approach policy strategies that address access to vaccines among PWDs in Zambia.

and universal health coverage. does this issue study through this study? justify

Reviewer #3: Abstract: Abstract should be revised to a more concise format. The first two lines of the abstract do not contribute significantly to the paper.

Revise ‘determinants of vaccine uptake’ to ‘odds of vaccine uptake’

‘Safety of the vaccine’- statement is incomplete.

Introduction:

Line 52-59: These should be revised, as the paper focuses on uptake, with the assumption that vaccines are already available in the country studied, not on vaccine apartheid which is not relevant to the paper.

Lines 60-65: These should be moved from introduction to discussion.

Lines 71-73: Same comments as lines 60-65

There are other studies that have investigated the uptake of COVID-19 vaccines among other populations in Zambia and should be included to provide a better summary of the COVID-19 vaccine situation in Zambia. The introduction should be revised, and be written in amore concise manner, with a direct focus on the topic and outcomes of interest.

Methods:

Lines 120- 122: Define low uptake- There is either uptake or no uptake, not low uptake. Revisit the SAGE definition, as the definition provided is for vaccine hesitancy. It is not correct to assume that all participants who had not yet received a COVID-19 vaccine were hesitant, as the participants are a vulnerable group who would have significant limitations in accessing vaccine services.

Lines 126-128: It appears that the authors are conflating non-uptake, with willingness to receive a COVID-19 vaccine (unsure/undecided). It appears that there are two outcomes being investigated- uptake among the study population, and willingness to receive among those who had not yet received a COVID-19 vaccine. The authors should separate the outcomes, and treat them as two different outcomes.

Line 133: ‘participants randomly selected’. Insert ‘were’

Line 210: Same comments as in lines 126-128

Results:

Line 223: Include the proportion of male/female participants in the first lines of the results.

Table 1: Include the number and proportion of vaccinated and unvaccinated in Table 1. The sum of the cell percentages should add up to 100% based on either the row total or the column total for that variable, and not the total number of participants. E.g proportion of vaccinated participants aged 18-35 should either be 230/666 (number vaccinated as denominator- (34.5%)), or 230/375 (denominator is total participants aged 18-35 (61.3%)). Authors should do the same for all cells.

Table 1: Income should either be reported in ZMW or USD, and be consistent in the result narrative and the table.

Fig 3: The text speaks to ‘Type of disability and the proportion of COVID-19 vaccine uptake’, however the title of Fig 3 reads ‘The proportion of Disability by sex among PWDs’. Please revise.

Line 259: Past behavior towards vaccination- data not included in table 1.

Line 272: Change ‘servility’ to severity.

Line 307: This should be table 2, table 4.

Table 2:

Is the dependent variable vaccine uptake or vaccine acceptance? A person may accept a vaccine but have not yet actually received the vaccine (uptake).

Row not aligned

Why were variables not significant at 5% level of significance included in the unadjusted logistic regression model included in the adjusted model (e.g extent of disability, vision, hearing, speech, infected with COVID-19 before)? I suggest they are removed from the unadjusted model.

Discussion:

Line 346: What are the determinants of COVID vaccine uptake in the general population, and how do they vary from those in the study?

Line 347: There are no results for vaccine acceptance. The authors should be consistent with vaccine uptake.

**Do you want your identity to be public for this peer review?** For information about this choice, including consent withdrawal, please see our Privacy Policy

Reviewer #1: No

Reviewer #2: **Yes: ** Vijay Kumar khanal

Reviewer #3: No

---

## [Decision Letter · Decision Letter 1]

PGPH-D-24-01701R1

Determinants of COVID-19 Vaccine Uptake Among People with Disabilities  in Three Selected Districts of Zambia

Dear Mwiinde,

Thank you for submitting your manuscript to PLOS Global Public Health. After careful consideration, we feel that it has merit but does not fully meet PLOS Global Public Health’s publication criteria as it currently stands. Therefore, we invite you to submit a revised version of the manuscript that addresses the points raised during the review process.

We look forward to receiving your revised manuscript.

Kind regards,

Collins Otieno Asweto, PhD

Academic Editor

Journal Requirements:

Additional Editor Comments (if provided):

Reviewer's Responses to Questions

**Comments to the Author**

Reviewer #4: (No Response)

Reviewer #5: (No Response)

Reviewer #6: (No Response)

Reviewer #7: (No Response)

publication criteria?

Reviewer #4: Partly

Reviewer #5: Partly

Reviewer #6: Partly

Reviewer #7: Yes

3. Has the statistical analysis been performed appropriately and rigorously?

Reviewer #4: Yes

Reviewer #5: Yes

Reviewer #6: I don't know

Reviewer #7: No

4. Have the authors made all data underlying the findings in their manuscript fully available (please refer to the Data Availability Statement at the start of the manuscript PDF file)?

Reviewer #4: Yes

Reviewer #5: Yes

Reviewer #6: Yes

Reviewer #7: Yes

5. Is the manuscript presented in an intelligible fashion and written in standard English?

Reviewer #4: No

Reviewer #5: No

Reviewer #6: No

Reviewer #7: Yes

Reviewer #4: The authors describe an analysis of determinants for COVID-19 vaccine uptake among people with disabilities in Zambia. The manuscript is useful for comparison to countries with similar characteristics and to assist researchers in developing credible methodologies for addressing similar research questions in their own countries. This is my first review of this manuscript. The authors have provided responses to previous reviewers and have made major changes.

However, the manuscript still requires significant revisions. I have gone through the revised version and made suggested revisions using the attached line numbering. Many of these suggested revisions involve mis-spellings, poor sentence construction, and confusing analyses. Even with my suggested revisions, it would be helpful if the authors could involve a professional copy editor in order to ensure that the next version of the manuscript is coherent and publishable. Good luck with this endeavor!!

SPECIFIC COMMENTS (Using the line numbering on the revised manuscript)

ABSTRACT: (This version probably exceeds the recommended length for an ABSTRACT. It may be appropriate to remove the AOR, CI, and p values. )

Lines 16-18: I would suggest starting this paragraph with " The WHO have declared that COVID-19 no longer constitutes a public health ememgency of international concern. However, it is still a public health threat due to uncertainties around the potential evolution of the virus, changes in clinical characteristics, and the introduction of new prevention and therapeutic modalities."

Lines 26-30. Rewrite the sentence, as "Determinants of higher vaccine uptake were: (1) married, ......." I would suggest that you do not capitalize each determinant, and remove the periods between each determinant.

MAIN TEXT:

Lines 41 46: The first paragraph is extremely confusing. These global trend numbers seem to overlap. Please rewrite the entire paragraph.

Line 49: rewrite as follows: "Although a person can still become infected with COVID-19 after vaccination...."

Line 54- 56, rewrite as follows: "Currently, there are growing concerns about vaccine uptake, which are auto challenges in access, high cost of production and purchase price, and high levels of misinformation (infodemic) on th effectiveness and safety of the vaccines. "

Line 57: place a comma after pandemic.

Line 63-65: Spell out "Persons with Disabilities" rather using PWDs, since this is the first time that the abbreviation is used. Please rewrite this paragraph. it is confusing.

Line 68: Place "health belief model" in quotations.

Line 82:, remove the phrase"...which was crucial for protection."

Lines 90-93: Rewrite as follows: "In the adult population, studies that focused on vaccine acceptance estimated the coverage to be 72% among health care workers, while in the general population acceptance was low in the early days of the pandemic"

Lines 93-94, rewrite as follows: "The acceptability of COVID-19 vaccination among parents to have their children vaccinated was high, but they indicated hesitancy about receiving the vaccine themselves."

Line 107: This would be the appropriate place for a full definition of PWD in Zambia.

Line 113: Change to "Operational Definitions"

Line 118-119: Revise sentence to "Vaccine hesitancy was defined as the refusal or denial of uptake of the vaccine when it was readily available, as defined by the Strategic Advisory Group of Experts(SAGE) on Immunization."

Line 123-124: Revise to read, "The second outcome variable of interest was no uptake of COVID-19 vaccine. This is further defined as respondents did not take a single dose of COVID-19."

Lines 133 to 137 do not make sense and need to be revised.

Line 139: Remove the phrase"..of the participants..." from the sentence

Lines 173 to 176. I don't understand what the phrase is trying to explain, "The theoretical structure or relevance of the health belief model" Is this a label for a FLigure???

Lines 184 to 186: Its not clear what this phrase means. "The test was to assess the validity and ambiguous statement of the...."

Lines 245 to 246: These two sentences do not make sense. Could this be rewritten?

Line 264: rewrite this section of the sentence as follows , "......particpants indicated yes about being aware of or had heard of the COVID-19..."

Lines 280 to 286 need to be rewritten. Most of this paragraph does not contain complete sentences.

Lines 288 to 296: This entire paragraph needs to be rewritten.

Lines 312 to 318: This entire paragraph needs to be rewritten.

Line 351: Replace the word "head" with "herd" and remove the word, "...this.."after the word "PWD"

Lines 353 to 354: The sentence beginning with, "However......."needs to be revised. Currently it is confusing.

Lines 358 to 359: I would suggest removing the sentence, "The prevalence of vaccine acceptancy

(acceptance)..." can be removed. It is repetitive.

Line 362, replace the word "...high..." with "higher"

Lines 372 to 373: Can you rewrite this sentence so that it is more clear?

Line 376 to 378, Can you rewrite the sentence, "This also can be attributed to the servility levels of disability......" The meaning is not clear.

Line 380: replace the word, 'brails" with "..instruction leaflets in Braille.."

Lines 388 to 390: Rewrite the sentence as follows, "This is different from the findings of Myers et al [49] who indicated that in the USA, PWDs were only concerned with the issue of safety of the vaccine."

Lines 412 to 415: This should be two separate sentences, as follows: "Those who perceived that COVID-19 had a moderate risk to members of the public were associated with the likelihood of less uptake. This is most likely due to COVID-19 conspiracy theories that have demonstrated.........."

Lines 415 to 417: Can you rewrite this sentence so that it is less confusing? "Those who disagreed that they......."

Lines 424 to 426, rewrite the sentence that begins, "Among those with Primary education......." It is confusing.

Lines 435 to 436, in this sentence replace the word, "...chocks.." with "shocks"

Lines 449 to 452: This sentence is confusing and should be revised.

Lines 455 to 458. Please revise this sentence. Its meaning is unclear.

Line 468. Replace the word, "dearth" with "death"

LIne 469. Replace the word "Improve..." with "Improved....."

Reviewer #5: See attached.

Reviewer #6: The study used the health belief model to help understand the underpinning determinants in accessing COVID-19 vaccines among PWDs in three selected districts of Zambia. The authors can further improve the manuscript by considering these suggestions.

ABSTRACT

95% CI should be written like this where applicable in abstract and text (e.g. table 2.) Also in this example and all others, 67.6% (95%, CI: 65% – 71%), 65% – 71% should be expressed in decimals and the same number of decimal points maintained throughout abstract and text. Lines 25 -30, must be well written making good use of punctuation and completing sentences. Authors should consider deleting lines 31 -33 because of the repletion of too many numbers in the abstract and concentrate on the key findings.

Introduction

Line 47, “Vaccines are key pharmaceutical interventions to contain COVID-19…….” Vaccines go beyond COVID-19. Give a general definition of vaccines to precede Line 47 to situate it in context and carry the reader along.

Line 53, replace pandemic with disease.

Line 85, improve vaccination coverage should be improve vaccination coverage and uptake

Line 90, state the other population. Line 91, give the range (in years) of the adult population. Line 92, is Vaccine acceptance referring to Vaccine uptake in this context? If so, use Vaccine uptake to maintain consistency and carry the reader along. Line 93, “general population acceptance was law…….”acceptance should be vaccine uptake if used interchangeably with Vaccine acceptance and law should be low.

Lines 93- 95, “The parents’ acceptability of COVID-19 vaccination was high for their children, however, they indicated high levels of hesitancy among themselves [30]”. Precede with Per vaccine uptake, parents’ acceptability of COVID-19 vaccination was high for……………………….

Lines 66- 71, move it to replace Lines 96- 97 while maintaining “in three selected districts of Zambia” at the end.

Line 99, Study Design and Study Area and Study Population should be Study Design, Study Area and Study Population

The study targeted PWDs aged 18 years and above. Any reason?

Line 113, delete local context.

Line 124, …..”no uptake this means that the respondents did not take any single dose of the COVID-19 vaccine (unsure, or undecided) as it was available”. Which is which relative to unsure, or undecided? Specificity is key to match the operational definitions.

Line 138 - 140 “During the contact period, if the participants could not be reached or their households were not identified, the next available name of the participants was selected from the sampling list and visited.” How? Explain well or delete

Line 140 - 141 “From each household, only one participant who was the eldest was included in the study even when they had two or more PWDs”. Is that not bias, explain why?

Line 151 precede vaccination with COVID-19

Apart from self-efficacy in Line 168 - 172, Lines 176 - 182 are repetitive. Delete one of these (lines) but maintain the reference in Line 176 - 182.

Results

Under results, give the subheading Demographic Characteristics of participants (PWDs) to start the description of results

There should be consistency in the reporting of the percentages. It cannot be 50% and 37.9%. Keep to one style throughout the manuscript. Also provide actual numbers to the percentages.

Table 1, what does the superscripts for the p value and the other variable (cash and other income) means? Indicate as a footnote. Indicate the percentage for the unvaccinated and use “n” not “N” for the vaccinated and “N” for the total. Take a detailed look at ALL the number and correct where applicable. How is it that Unemployed, 672 is greater than the total of 666 under vaccinated? Reanalyze the data where you have to.

Line 233 -235 should be added to the write up on the results and the fig 2 indicated deleted.

Line 240, Fig 3 should be Fig 2. Fig 2 – label the other coloured legend as done for the blue

How different is the vision from invisible? Were you participants not PWDs? Line 237 -240, Take a detailed look at all your percentages. At a cursory look, if 666 participants were vaccinated, 555 of such participants with movement disability should give 83.3 % and not 51%. Check for all. You may have to reconsider a new figure with the right numbers to describe the write up

Proportion of Disability by Sex among PWDs

Line245, “A total of 75.13% (n=740) had acquired disability of this number males had 365”. This is very confusing. All along I was of the view you sampled 985 PWDs and not 740. What happened to the difference of 245 and how will you categorized them? In any case, fig 3 and not fig 4 as indicated in Line 248 and 249 has no record of “others” in the axis labeled “Type of Disabilities of Participants by sex”. Also I expected males to be 489 by inference from line 217 and not 365. Could the authors explain the inconsistencies? You may have to reconsider a new figure with the right numbers to describe the write up. Fig 3 - How different is vision from invisible and were your participants not PWDs? Why do you single out Congenital disabilities in lines 245-246 and is that to say vision, hearing amongst others indicated in fig 3 could not have been congenital? Which of these disabilities are congenital as against non-congenital?

Line 255, (CI; 68.99-74.60)- indicate the level or percentage of CI.

Lines 253 – 261, For Past Behaviour toward Vaccination, indicate the actual number out of the total and the percentages. For example (250/ 550, 45.5%)

Line 270, Perceived Severity of COVID-19 – while susceptibility has to do with risk (low, high etc), severity has to do with progression of the risk, whether asymptomatic or symptomatic with the infection leading to other secondary infections or degenerating to diseased stage leading to death. The write up doesn’t address the theme (Perceived Severity). Comment rather on the PWDs response to hospitalization and death. Refer to S1 Table

Line 290 – 291 while 27.97% (n=273). Complete the sentence

Ques to Action. How different is this from lines 170, 179 (Cues to action: Internal or external stimuli that trigger the decision-making process to take action) and is Ques as in question? Authors have to rewrite the section again to suit S1 Table.

Lines 326, (AOR= 10.58, 95%, CI: (1.56-72, p= 0.025) should be ………….95% CI…………..- 72.00

Discussion

What were the drivers or metrics informing Past Behaviour toward Vaccination

Conclusion – conclude by stating key findings of study

S1 Table, indicate the total number and represent with “N”. For example N=985.

Reviewer #7: 1 Title OK

2 Abstract OK

3 Background OK

4 Methods The following need to be considered.

i. Exclusion criteria. One of your exclusion criteria is to exclude those who did not give consent to participate in the study. If that was the case, then you would have ending up with 100% response rate. How can you assess for non-response bias?

ii. Study outcomes are confusing. If you considered the multivariate binary logistic regression model then your dependent variable MUST be dichotomous with categorical or discrete outcome value. Its either someone has ever taken a COVID 19 vaccine or not. If your dependent variable is COVID 19 vaccine uptake, then your outcomes should be either YES or NO.

iii. Estimating sample size. Sometimes it is not necessary to have a local prevalence rate. If you don’t have a country prevalence you may consider the available prevalence from other countries within a region or continent or global rather than using a default rate of 50% for an unknown prevalence rate. Considering my above comment on response rate, I wouldn’t have expected you to adjust for non-response while you excluded all those who did not consented to participate in the study.

iv. The COVID 19 Vaccine uptake rate was found to be high 67.6%. The odds ratio can estimate risk for rare conditions/events. For common conditions/events it may lead to overestimation point and interval estimates. The correct measure would have been prevalence ratio using the log binomial regression model. You may read more from these references.

a. Barros AJ, Hirakata VN. Alternatives for logistic regression in cross-sectional studies: an empirical comparison of models that directly estimate the prevalence ratio. BMC Med Res Methodol. 2003 Oct 20;3:21. doi: 10.1186/1471-2288-3-21. PMID: 14567763; PMCID: PMC521200.

b. Coutinho LM, Scazufca M, Menezes PR. Methods for estimating prevalence ratios in cross-sectional studies. Rev Saude Publica. 2008 Dec;42(6):992-8. English, Portuguese. PMID: 19009156.

5 Results Depends on re-analysis

6 Discussion Depends on re-analysis

7 Conclusion Depends on re-analysis.

**Do you want your identity to be public for this peer review?** For information about this choice, including consent withdrawal, please see our Privacy Policy

Reviewer #4: **Yes: ** Paul De Lay, MD, DTM&H (Lond)

Reviewer #5: No

Reviewer #6: No

Reviewer #7: No

---

## [Decision Letter · Decision Letter 2]

PGPH-D-24-01701R2

Determinants of COVID-19 Vaccine Uptake Among Persons with Disabilities  in Three Selected Districts of Zambia

Dear Allan,

Thank you for submitting your manuscript to PLOS Global Public Health. After careful consideration, we feel that it has merit but does not fully meet PLOS Global Public Health’s publication criteria as it currently stands. Therefore, we invite you to submit a revised version of the manuscript that addresses the points raised during the review process.

We look forward to receiving your revised manuscript.

Kind regards,

Collins Otieno Asweto, PhD

Academic Editor

Journal Requirements:

Additional Editor Comments (if provided):

**Comments to the Author**

Reviewer #4: (No Response)

Reviewer #6: All comments have been addressed

publication criteria?

Reviewer #4: Partly

Reviewer #6: Yes

3. Has the statistical analysis been performed appropriately and rigorously?

Reviewer #4: Yes

Reviewer #6: (No Response)

4. Have the authors made all data underlying the findings in their manuscript fully available (please refer to the Data Availability Statement at the start of the manuscript PDF file)?

Reviewer #4: Yes

Reviewer #6: Yes

5. Is the manuscript presented in an intelligible fashion and written in standard English?

Reviewer #4: No

Reviewer #6: No

Reviewer #4: This is my second review of this manuscript. Most of the revisions that I suggested previously were incorporated, but not all. The article still needs considerable proof-reading. There are multiple mis-spellings, poor sentence construction, confusing arguments, and counter-intuitive statements that are neither discussed nor explained.

I am using the Line numbering from Revision 2 with Track Changes turned off.

ABSTRACT: The revised abstract is fine.

MANUSCRIPT:

Line 47, rewrite as follows, "With the current ongoing infections of COVID-19, vaccines.........."

Line 49, please spell "....illness.."

Line 51, I would suggest removing the phrase "Since there is no proven antiviral medication available for COVID=19..." and start the sentence as, "Most infected patients are managed through supportive care."

Line 55. Rewrite the sentence as, "Currently there are growing concerns about vaccine uptake, which are due to challenges in access,...."

Line 62. I would delete this entire sentence. "This situation may result in a better......."

Line 65. Please rewrite as follows: "Persons with Disabilities (PWD) face challenges in accessing primary health care (PHC) services, including access to vaccination, compared to the general population. (12) This is particularly true for those residing in rural areas and are PWDs.

Lines 75-76, please revise as follows: "In August 2021, the vaccine was rolled out...."

Lines 88-89, please rewrite as follows. "Parent's acceptance for COVID-19 vaccination was high for their children,..."

Line 94.please rewrite as, "....healthcare workers, low-income setting areas (slums) university students......."

Lines 98-99, please rewrite as, "In the past, vaccine hesitancy was high in Africa for COVID-19 and for other vaccines, like polio vaccine. If not addressed this is likely to sabotage the vaccination gains made already.

Lines 108 to 110. Delete this sentence. It is repeated in the next paragraph under Inclusion Criteria.

Lines 112-114. please rewrite as follows, "PWDs who are 18 years and above are registered with the Social Welfare Department. Additionally, PWDs who were not on the social welfare register were included if they had an approved medical record of disability from a recognized government health institution."

Lines 159-160. Please rewrite as follows, "In this present study, the HBM was adopted as a conceptual framework to assess PWD's determinants for vaccine uptake based on identified literature variables of interest. "

Line 194, please place a colon (:) after "...to the following: age, sex......"

Line 239 rewrite the title as "Proportion of Disability by Sex and Location among PWDs"

Line 240. Please put a period after the word disability and then start a new sentence.

Lines 244-245. This sentence does not make sense.

Lines 262-265. This sentence is counter-intuitive. Can the authors rewrite this or better explain this finding.

Lines 267 to 272. Same as above. Can authors better explain this finding?

Lines 277 and 278. Please replace the words "dearth" with the word "death"

Line 295. Please rewrite this section of the sentence, ".....having the ability to visit PHC on their own...." Not "own their own"

Line 323, please replace the word, "windowed" with "widowed"

Line 382. Replace the word, "servility" with "severity"

Line 393, replace the word, "windowed" with "widowed"

Line 400 rewrite as follows, ".....platforms which has caused less uptake among members of the public."

Line 429, please rewrite as follows: "...infected by COVID-19 in THE future......

Line 441. please rewrite as follows, "....counries, PWD reported worse access to health care, due to transportation costs and distance to the primary health facilities."

Line 451, replace the words "...archiving head..." with "...achieving herd immunity..."

Lines 477-478, rewrite as follows:"Special messages targeting PWDs should be developed to ensure that they are not left behind."

Reviewer #6: (No Response)

**Do you want your identity to be public for this peer review?** For information about this choice, including consent withdrawal, please see our Privacy Policy

Reviewer #4: **Yes: ** Paul R De Lay, MD, DTM&H (Lond)

Reviewer #6: No

---

## [Decision Letter · Decision Letter 3]

PGPH-D-24-01701R3

Determinants of COVID-19 Vaccine Uptake Among Persons with Disabilities  in Three Selected Districts of Zambia

Dear Dr. Mwiinde,

Thank you for submitting your manuscript to PLOS Global Public Health. After careful consideration, we feel that it has merit but does not fully meet PLOS Global Public Health’s publication criteria as it currently stands. Therefore, we invite you to submit a revised version of the manuscript that addresses the points raised during the review process.

We look forward to receiving your revised manuscript.

Kind regards,

Miquel Vall-llosera Camps, Ph.D.

Staff Editor

Journal Requirements:

Reviewers' comments:

Reviewer's Responses to Questions

**Comments to the Author**

Reviewer #4: All comments have been addressed

publication criteria?

Reviewer #4: Yes

3. Has the statistical analysis been performed appropriately and rigorously?

Reviewer #4: Yes

4. Have the authors made all data underlying the findings in their manuscript fully available (please refer to the Data Availability Statement at the start of the manuscript PDF file)?

Reviewer #4: Yes

5. Is the manuscript presented in an intelligible fashion and written in standard English?

Reviewer #4: Yes

Reviewer #4: This is my second review of this manuscript. The authors have responded to all of my suggestions. The manuscript is now more readable and the counterintuitive findings are more clearly explained. There are still a few minor revisions which are outlined below. Once these revisions are made I feel that the article is ready for publication.

Line 75: Place a period at the end of the sentence, "....and mobile vaccination points."

Line 77: Place a comma as noted. "....vaccination coverage and uptake, the government of the Republic...."

Line 199, replace the word, "got" with "received"

Lines 346 to 347. I would suggest deleting the first sentence in the DISCUSSION. "This study examined the determinants of COVID-19 vaccination uptake among PWD in selected districts in Zambia." You essentially repeat all of this in the second sentence.

Line 351 to 352. I would suggest rewriting the sentence as follows (revisions are IN CAPS), "The low levels of vaccination among PWDs in Zambia follow a similar pattern to OTHER sub-Saharan African countries BY not reaching the required herd immunity of 80%."

Line 359, place a comma after the word "outcomes"

Line 392, place the word "an" before the word "infodemic"

Line 409, place the word, "in" before "adulthood"

Line 410 to 412. Please rewrite this entire sentence. It doesn't make any sense. "This is an indication that to improve vaccination among PWDs there is a need to ensure that inclusive vaccination coverage to improve uptake during pandemics such as COVID-19."

**Do you want your identity to be public for this peer review?** For information about this choice, including consent withdrawal, please see our Privacy Policy

Reviewer #4: **Yes: ** Paul R DeLay, MD, DTM&H (Lond)

---

## [Editor Report · Decision Letter 4]

Determinants of COVID-19 Vaccine Uptake Among Persons with Disabilities  in Three Selected Districts of Zambia

PGPH-D-24-01701R4

Dear Mr Mwiinde,

We are pleased to inform you that your manuscript 'Determinants of COVID-19 Vaccine Uptake Among Persons with Disabilities  in Three Selected Districts of Zambia' has been provisionally accepted for publication in PLOS Global Public Health.

Best regards,

Julia Robinson

Executive Editor